# Template-Assisted Co-Ni Nanowire Arrays

**DOI:** 10.3390/nano9101446

**Published:** 2019-10-11

**Authors:** Ruxandra Vidu, Andra M. Predescu, Ecaterina Matei, Andrei Berbecaru, Cristian Pantilimon, Claudia Dragan, Cristian Predescu

**Affiliations:** 1University Politehnica of Bucharest, Splaiul Independentei nr. 313, Bucharest, sector 6, CP 060042 Bucharest, Romania; rvidu@ucdavis.edu (R.V.); ecaterinamatei@yahoo.com (E.M.); andrei_berbecaru@yahoo.com (A.B.); cristi_pantilimon@yahoo.com (C.P.); draganclaudiaa@yahoo.com (C.D.); predescu@ecomet.pub.ro (C.P.); 2Department of Electrical and Computer Engineering, University of California Davis, One Shields Avenue, Davis, CA 95616, USA

**Keywords:** Co-Ni alloy, anomalous deposition, thin films, template-assisted electrodeposition, nanowire arrays

## Abstract

A comparison was performed between Co-Ni thin films and template-assisted nanowires arrays obtained by electrochemical co-deposition. To reduce the effects of anomalous deposition and increase the Ni content in the deposit, an electrolyte with three times more Ni than Co in atomic ratio was chosen. Electrochemical deposition was performed at constant potentials chosen in the range from *E* = −0.8 to −1.2 V vs. Ag/AgCl. Cyclic voltammetry, chronoamperometry, and charge stripping techniques were used to characterize and compare the electrochemical behavior of Co-Ni films and nanowires. Morphological and compositional characterization was performed by scanning electron microscopy (SEM/EDAX) to assess the influence of the deposition potential on the growth of film and nanowires. A comprehensive analysis of the deposit growth rates for thin films and nanowires is presented taking into consideration the hydrogen evolution and anomalous deposition. The comparative study of the composition of film and nanowires obtained at different deposition potentials has shown that deposition of nanowires with a film-like composition takes place at more positive potential than thin film.

## 1. Introduction

In search of material systems with tunable magnetic properties, Co-Ni alloys present an interesting combination of low and high magnetocrystalline anisotropy of Ni and Co, respectively. Electrochemical deposition of cobalt and nickel alloys has been studied due to their known applications in the computer industry, recording devices, and magnetic memories [1,2,3,4,5]. Cobalt-nickel alloy forms a solid solution over the whole concentration range, which enables to modulate the magnetic properties according to the cobalt and nickel percentage in the alloy. Although miniaturization has triggered additional challenges such as nanodevice integration, electrochemical templated synthesis of nanowires has provided a cost-effective solution for integration of nanowires into devices. When the Co-Ni alloy is obtained in the form of nanowires, their intrinsic high aspect ratio has a direct impact on their magnetic, electrical, and mechanical properties. Due to their magnetic properties individual nanowires or nanowire arrays are currently being studied [6,7].

Nanostructured magnetic materials with certain magnetic behavior, such as Co-Ni thin films and nanowires, have been studied primarily to control the magnetic and magnetotransport performance of nanostructured systems and devices [5,8,9,10,11]. The differences between thin films and nanowires include the motion of the magnetic domain wall and the pinning sites such as the interface between segments in a multisegmented nanowire can control the spacing between domain walls. In the array format, Co-Ni nanowires show magnetic interactions that are completely different from the individual Co or Ni nanowire arrays [12]. Magnetostatic dipolar interactions between nanowires can dictate the direction of easy magnetization, i.e., either perpendicular or parallel to the NWs (nanowires) growth axis [13,14]. More recently, different research groups have reported on the template-assisted electrochemical deposition of Co and Ni-rich segmented nanowires, for which different magnetic behaviors have been observed depending on the composition and size of the segments [5,10,11,12,15,16]. The production of Co-Ni segmented nanowires in the pores of a membrane requires the control of deposition parameters to change the composition from one segment to another. When electrodeposition is performed at a given potential in a single bath, nanowires of a particular composition can be obtained [5]. Pereira et al. [11] have obtained multisegmented Co-Ni nanowire arrays containing from 3.38% to 97.28% Ni using five successive deposition baths containing from 0% to 100% Ni.

Using a single deposition bath, multisegmented, multi-composition, nanowires can be obtained electrochemically by varying the applied potential over a certain period of time for each segment. For example, multisegmented Co-Ni nanowire arrays with alternating segment compositions of Co54Ni46 and Co85Ni15 have been obtained by electrochemical deposition in the pores of anodized alumina membrane [10,15]. However, the co-deposition process and the composition of each segment are more difficult to control in the Co-Ni system due to the anomalous deposition, where Co is preferentially deposited over Ni [17,18,19]. Although the use of a single deposition bath to grow multisegmented nanowires array has clear economic advantages over multiple baths, Co-rich and Ni-rich multisegmented nanowires have not yet been obtained in a single bath. 

In order to design advanced Co-Ni magnetic nano-systems of multisegmented nanowire arrays, extensive research is needed to better control the composition, structure, and size of nanowires with Co-rich and Ni-rich alternating segments. In this paper, a comparative study was performed between thin films and nanowires of Co-Ni alloy obtained by co-deposition from a single bath. To reduce the effects of anomalous deposition and increase Ni in the deposit, we used an electrolyte containing three times more Ni than Co in atomic ratio. Since a simple extrapolation of film electrodeposition parameters to the template-assisted NWs array will not work, this comparative study aims to better understand how the deposition parameters affect the composition and the morphology of films and nanowires. The comparison between film and nanowires was performed using Au-sputtered PCTE membrane as electrode, which was set up in two different configurations to allow for either film on nanostructured Au or nanowires deposition.

## 2. Materials and Methods

Solutions containing cobalt and nickel were prepared by dissolving 30 g/L NiSO_4_∙6H_2_O and 10 g/L CoSO_4_∙6H_2_O in aqueous solutions containing 40 g/L H_3_BO_3_. All chemicals were purchased from Sigma-Aldrich (St. Louis, MO, USA). Deionized water (Milli Q l8-MX, Kenilworth, Merck. NJ, USA) was used for preparing solutions and for rinsing. 

Sample electrodes were constructed from track-etched polycarbonate (Sterlitech, Inc., Kent, WA, USA) with an average pore size of 200 nm and thickness of 7 μm. Membranes were metallized with Au via sputter deposition to achieve an electrode with a thickness of approximately 50 nm. A copper tape current collector with Ni-free conductive adhesive was attached to the Au side of the membrane, and the membrane was encapsulated in plastic tape (Brother Corporation) except for a circular cut-out area of a 0.3846 cm^2^ hole exposing the membrane surface. For film deposition the electrode was constructed with the Au electrode exposed to the solution; for nanowire deposition the non-coated side was exposed to solution, forcing deposition to proceed within the pores. An illustration of the sample setup is shown in Figure 1. The sample electrodes arranged in nanowire configuration were sonicated for 5 min before each experiment in the solution to be used to allow for the solution to enter the pores. 

Electrochemical pre-treatment of the electrode: an electrochemical treatment of the Au surface was performed in the beginning of each experiment in order to ensure same surface conditions before the electrodeposition experiment. This treatment also greatly increases the reproducibility of the results. The electrochemical treatment of Au in 50 mM H_2_SO_4_ and the surface processes that led to a smooth Au surface are detailed elsewhere [20,21]. 

Electrochemical experiments were performed in a conventional three-electrode setup including a potentiostat/galvanostat PARSTAT 4000 (Princeton Applied Research AMETEK, Berwin, PA. USA). The counter electrode was made of Pt and the reference electrode was Ag/AgCl electrode (3 M NaCl). The potentials presented here are relative to Ag/AgCl 0.194 V vs. Standard Hydrogen Electrode (SHE). All potentials given in this paper are given in reference to Ag/AgCl unless otherwise specified. The Versa Studio software was used to control the computer. Deposition of Co-Ni films was performed at room temperature at constant potentials *E* = −0.8, −0.9, −1.0, −1.1, −1.2 V for 15 min. For nanowires, the deposition time was adjusted to obtain either nanowires or overgrown nanowires caps. Electrochemical characterization was performed by cyclic voltammetry, chronoamperometry, and charge stripping.

Structural and morphological characterization of nanowires was performed by scanning electron microscopy (SEM) equipped with an X-ray dispersive energy (EDS). To expose the nanowires array, the PCTE membrane was dissolved in a concentrated Cl_2_CH_2_ solution and then washed with Milli-Q water and dried with N_2_.

## 3. Results and Discussion

### 3.1. Electrochemical Characterization Co-Ni Thin Films and Nanowires

Cyclic voltammetry (CV) experiments were performed from 0.7 V to a vertex potential of up to −1.2 V at 20 mV/s. Figure 2 shows the typical cyclic voltammetry scan of Au in the Co-Ni solution for both film and nanowires configurations. 

Single-element deposition of Co and Ni are both well understood for both metallic films. The standard potential for the reduction half reaction of Co and Ni are very close to each other, i.e., −0.28 and −0.25 V vs. SHE, respectively [22]. The standard potential for co-deposition of Co and Ni is −0.53 V vs. SHE (or −0.331 V vs. Ag/AgCl) which correspond to the potential at which the onset of the current increase was observed in the CV scan of the film in Figure 2. Going to more negative potentials, the large increase in current is associated with the hydrogen evolution reaction (HER) overlapping with Co-Ni co-deposition. Hydrogen reduction and evolution from the Co-Ni film is higher compared to nanowires, mainly due to the large surface area of the film. Hydrogen evolution is a multi-step reaction process that inevitably occurs simultaneously with the electro-chemical deposition at negative potentials and the hydrogen deposition current on the Co-Ni alloy surface increases with increasing deposition current at higher overpotentials. The desorption of the adsorbed hydrogen, H_ads_, is not seen in the CV due to the high surface diffusion of H_ads_ and the formation of H_2_ (gas) that form small bubbles and eventually evolve from the surface. In electrochemical deposition, hydrogen evolution affects the morphology of the growth surface [23]. Gas bubbles formed on the surface were visually observed at the membrane surface during routine experimental work. 

When the potential is reversed, the scan crossovers the initial scan at –0.4 V, which correspond to point A in Figure 2. This crossover in CV is a signature of the nucleation mechanism in depositing Co-Ni alloy on Au. Then, when the potential is scanned further in the positive direction, two peaks B and C can be observed at 0.15 and 0.3 V respectively, corresponding to Co and Ni dissolution. The last layers deposited are the first one to desorb. For the NWs configuration electrode, the two peaks are smaller and shifted to more positive potentials compared to film, which indicate a more difficult oxidation mechanism of Co-Ni from nanopores compared to film. The inlet in Figure 2 clearly shows that the desorption peaks in the NW configuration electrode are shifted by more than 60 mV in the positive direction compared to the film.

### 3.2. Electrochemical Deposition

Co-Ni deposition on and into polycarbonate membrane was performed in approximately 150 mL of solution for periods of time ranging from 5 to 15 min at room temperature. A typical deposition experiment observed the following procedure: first, two CV cycles were performed from 0.7 to −1.2 V followed by two CV cycles from −0.7 to −1.4 V, at a scan rate of 20 mV/s. After cycling, the OCP was measured for 3 s and then deposition was performed at constant potential E, where *E* = −0.8, −0.9, −1.0, −1.1, and −1.2 V vs. Ag/AgCl.

Deposition time was different for the two electrode configurations. From our experience, a uniform film of Co-Ni is deposited in 10–15 min, while the nanowires grow inside the PCTE membrane in less than 5 min. To avoid the interference of overgrown nanowires in the calculation of the growth rate, we compared the charge associated with the 5 min deposition time on both sample configurations. The charge associated with the deposition at a certain potential, *Q(E)*, was estimated as the area under the curve, *i* = *f*(*t*), according to the following equation:(1)$$Q\left(E\right)=\overset{t=900s}{\underset{t=1s}{\sum }}i\;dt,$$
where *Q* is the charge, C; *i* the recorded current, A, and *t* is the time, s. 

When comparing the charge associated with the deposition of film and NWs (Figure 3), we noticed that the variation of the NWs growth rate with the deposition potential is higher than the film. The increase in nanowire deposition rate when the deposition potential move to more negative values exceeds the growth rate of the film around −1.0 V. As the deposition rate also increases when overpotential increases, the nanowires can grow faster at potentials more negative than −1.1 V, where the two slopes intersect.

On the surface of a film, it is found that the H_2_ bubbles inhibit the electrochemical reaction underneath, blocking the direct contact of the electrode with the electrolyte [24]. Hydrogen evolution during the electrochemical co-deposition of two or more elements at high overpotential results in non-homogenous surface film [20,24,25]. In the NW electrode configuration, Au is found only on the pore wall closed to the surface, i.e., in a small area around the mouth of the nanopore. Therefore, hydrogen evolution that accompanies the electrochemical co-deposition of Co and Ni is slower in the NW electrode configuration than on the film, as can be also observed in the CV scans presented in Figure 2. This observation, along with the specificity of the electrochemical deposition inside the nanopores, may explain the increase in nanowires growth rate compared to the film. 

### 3.3. Morphological Characterization of Films and Nanowires

The morphology and the composition of films and nanowires were analyzed by SEM and EDS. Electrodeposition was performed at potentials between −0.8 and −1.2 V for 60–300 s for both film and nanowire sample configurations (Figure 4). The Co-Ni film surface (Figure 4a–c) is generally smooth with particle-like deposits uniformly “sprinkled” over the surface. The density of surface particles decreases with the increase in overpotential. The morphology of these deposits suggests the presence of Co-rich phase on the surface due to the dendritic appearance with plate like branches perpendicular to each other. Electrodeposited cobalt is known to have a nucleation and growth mechanism [26], which could be locally favored over Ni [27].

Typical SEM images of Co-Ni film, nanowire array, and overgrown nanowires film is presented in Figure 4d–f. Although the morphology of nanowires is similar at different deposition potentials, in the case of nanowires obtained at −1.2 V, several air packages are visible along nanowires (Figure 4f). During the hydrogen evolution, bubbles appear and adhere to the surface prior to evolving from the surface. Broken nanowires due to hydrogen reduction were observed in some pores at over-potentials where the current is also high due to the overlap of three reduction currents: Co, Ni, and H_2_. As far as nanowire growth is concerned, there is a competition between them, and those favored from the energy point of view are growing faster. Nucleation of new growth sites is more difficult than the process of continuous growth in a current nucleation site. In other words, those pores that have a good start in the nucleation will be filled at a faster rate than others. Once a nanowire has nucleated, it will begin to grow faster than others because the growth process is energetically more favorable. 

After the pores are filled with Co-Ni, the growth of the deposit is no longer restricted by the pore walls and thus an outgrowth cap is formed on the surface. When the nanowire reaches the surface, the enlarged surface area increases the total current flow of the electrodeposition, and the current variation over time resembles that of a film. Overgrown nanowires look a lot like mushrooms, hence the name associates with this morphology. Figure 4g–i shows the surface morphology of the film formed by the mushroom cups, which grow quickly laterally connecting the neighbor overgrown nanowires. 

The presence of the outgrowth caps on the sample surface suggests that the electrodeposition time was longer that required to fill the pores. If the growth of nanowires is perfectly uniform in all the membrane pores, the electrodeposition process will end before the caps begin to form. Figure 5 shows that the nanowires obtained at −1.2 V for 3 and 5 min are fairly uniform in height and filled the majority of the pores. The SEM images show a mixture of partially filled, completely filled pores and overgrown nanowires. This is mainly due to the fact that not all the pores have the same length, because not all are perpendicular to the surface. The increase in deposition time from 3 to 5 min increases the number of caps grown until all the surface is coated with of film of mushroom caps (Figure 5). 

In addition to the geometric factor, another explanation for the different growth rates of NWs in the same sample is the occurrence of localized reactions that occur unevenly in pores. Local overcurrent built in certain pores can lead to hydrogen reduction and evolution that compete with the reduction of Co and Ni ions. Similarly, hydrogen reduction can lead to bubble formation that will further reduce the available sites for Co or Ni nucleation and growth. This competition has been already observed in our work and on other surfaces [24,28]. Although, the formation of hydrogen bubbles can be reduced by mechanical stirring, there is little or no flow at the bottom of the pores. For those pores where the hydrogen bubble built up, there is a little chance to host long nanowires or to host nanowires at all. Another reason for unfilled nanowires is that the electrolyte does not reach the electrode, so there is no electrochemical reaction. To ensure that all the pores are open and filled with electrolyte, we sonicated the samples in the deposition solution for 3–5 min before each experiment.

As the growth of nanowires in the PCTE membrane is so erratic, the variation of current density with time is not a good method to establish when the nanowire reached the surface. In order to know when the nanowires reach the surface, the software should switch to a larger surface, but the membrane porosity cannot be used to calculate the current density, since not all the pores are open or open at once. As the nanowires reach the surface and outgrow vertically, the deposition area increases, which increase the current flow. As the surface transition from nanopores (~20% of the membrane surface) to outgrowth caps and then to a film is not taken into account by the software, the increase in current is actually displayed during deposition as an increase in the current density. Therefore, current-time recording cannot be interpreted in terms of current density for the nanowire growth.

### 3.4. On the Mechanism of the Electrodeposition

At negative potentials the growth of films and nanowires is fast (Figure 4 and Figure 5). Ebothe et al. [29] have found that Co-Ni interface follows a local conservative model. In experimental conditions that favor local processes, the growth of the interface is dominated by relaxation processes such as surface diffusion. In co-deposition, the competition of surface and interface processes can change the dynamics of local growths from conservative to non-conservative at high deposition rates [29]. High deposition rates deplete the cations in the double layers and the interface growth becomes controlled by the mass transfer in the electrolyte, electrochemical reaction, and surface diffusion.

On the other side, the overgrown nanowire cups follow a different growth mechanism. The overgrown nanowires are associated with the formation of the so-called mushroom caps that continue to grow until they merge and cover the entire surface of the membrane with a continuous film. Figure 6 illustrates the growth phases through which the mushroom cups increase in size during the electrochemical deposition, i.e., two growth mechanisms depending on the distance between them and deposition time. 

Initially, overgrown nanowires of similar size that do not have immediate neighbors could continue to grow independent caps (Figure 6a,b) which eventually would meet and develop sideways together as a single cap (Figure 6c). The mobility of the surface atoms is very high and is comparable with high temperature diffusion [30,31,32]. The electrochemical process by which small size cups tend to become smaller and eventually disappear while the growth continues on the respective larger cups is called electrochemical Ostwald ripening, a term that was introduced by Mulder et al. [33]. This process is similar to the conventional Ostwald ripening where the surface is minimized to reach the thermodynamic equilibrium. 

When caps meet, a parallel electronic circuit is formed that connects the cups with the connector and close the circuit. This gives rise to an intrinsic instability of the nano-caps surface in contact with the electrolyte because on one side, the corresponding ions are mobile, and on the other side, the metal ion transport is sluggish within material itself, which prevents the arrangement from undergoing conventional Ostwald ripening. However, the rapid metal ion transport through the electrolyte at large overpotentials results in the growth of the cup by a dual diffusional and electrochemical Ostwald ripening. Accordingly, the interplay of surface diffusion and electrochemical reactions determine the morphology of the caps.

### 3.5. Chemical Characterization of Films and Nanowires

SEM observation and EDAX were performed to correlate the composition of the deposit obtained at different deposition potentials. SEM study performed at a certain potential for different deposition times showed that the nanowires grow very rapidly. Since the current-time plots were not able to provide information on the exact moment when the NWs reach the membrane surface, we chose the stop the deposition time based according to the SEM observation. The deposition time was adjusted and decreased as the deposition potential was moved to more negative values. When the deposition time surpassed the nanowire growth, samples were covered with a layer that was characteristic of overgrown nanowires. As shown in Figure 7, the overgrown nanowires have a different morphology compared with the film obtained at the same deposition potential. While the film surface shows clear signs of uneven deposition, the outgrowth cap surface shows a more uniform composition on a large scale, but at the level of the individual cap, the top of the cap has more Co than between mushrooms. 

This aspect is more evident in Figure 8 where the surface morphology along with the EDS scan and elemental mapping of the overgrown layer of nanowires show distinct round shape morphology of the caps with a Co-rich top, which is consistent with the anomalous deposition theory [18]. In fact, deposition rate and surface diffusion dictate the chemistry and morphology of deposits on the electrode surface due to the development of competitive processes of nucleation and growth emerging during electrochemical deposition [3]. A balance of this competition is determined by the surface diffusion and the deposition flux. 

The morphology of the network structure of mushroom caps presented in Figure 8 is representative for all the overgrown NWs obtained in the deposition potential range from −0.8 to −1.2 V. The elemental mapping indicates that there is more Co than Ni on top of the mushroom caps than in between them. Additionally, the film obtained at −1.2 V shows on top of the mushroom caps some features specific to Co deposition. It is known that Co has a deposition mechanism based on nucleation and growth, which can be seen in the morphology of the film (Figure 4a–c). Morphology of the electrodeposited Co film is characterized by a branched dendrite surface with nanosized plate nanoparticles perpendicular to the dendritic branches (best seen in Figure 4g). Yanpeng et al. [34] have shown that the morphology of Co deposits is governed by the competition of two processes, a diffusion limited aggregation and the reaction kinetics at the growing deposit surface. In the atomistic approach of the formation of mushroom cups during deposition, Co atoms form hexagonal plate particles by surface diffusion [26].

Under applied electric potential, the growing interface is controlled by the concentration of the diffusing ions. In essence, electrodeposition leads to a non-local growth process since the electromigration and the diffusion of the species in the electrolyte imply the effect of the electric field. High deposition rates deplete the cations in the double layer and the growth of the deposit become diffusion controlled by the cation mass transfer through electrolyte. 

### 3.6. Ni Content in the Films and Nanowires

The composition of the Co-Ni deposit obtained at potentials between −0.8 and −1.2 V was investigated for both films and nanowires. Table 1 presents the composition of the deposit and Figure 9 shows the variation of Ni-Co ratio in the deposit at different deposition potentials. On the secondary y-axis, the variation of Ni in the deposit is displayed. These data were obtained from the EDAX measurements performed on respective samples deposited at different potentials.

Figure 9 shows that there is more Ni in NWs than in the film. This difference in composition between nanowires and films has been observed for other systems as well [28]. It is possible that Co deposition on a surface along with hydrogen evolution reaction delay the Ni deposition. Previous data obtained for Co-Sb [28] have shown that there was a clear difference in composition between a thin film and nanowires. Additionally, Figure 9 also shows that the deposition of a film with a nanowire-like composition takes place at more negative potential. 

Anomalous deposition of Co-Ni alloys in pores differs from the films largely because the growth medium inside a pore has no convection that keeps the solution near the interface in equilibrium with the bath. Surface reactions that are diffusion-sensitive show notable differences in a pore versus on a flat substrate, as seen in the unstable growth rate of nanowires within pores of differing lengths [24,28].

Ni content increases from 58 to 73 at.% in film and from 66 to 75 at.% in nanowires. The nanowires observed at −1.1 V had 76 at.% Ni in the composition of the deposited alloy, which is the highest content observed for Ni-rich deposit in a single bath. Prida et al. [15] obtained nanowire segments with maximum 46% Ni from a bath containing Ni:Co = 2.225.

## 4. Conclusions

This study presented the electrochemical co-deposition of Co and Ni, two metals that show anomalous deposition and compared the film and nanowires obtained in the potential range from −0.8 to −1.2 V. This comparative study between films and nanowires is important in the study of Co-Ni alloy nanowires with tunable properties. We have shown that there are several differences in electrochemical deposition of films and nanowire arrays from an electrolyte containing three times more Ni ions than Co ions, as follows: 

Deposition rate: the deposition rate increases as the deposition potential moves to more negative values. The variation of the NWs growth rate with the deposition potential is higher than the film. Nanowires grew faster than film at potentials more negative than −1.1 V. 

Hydrogen evolution: the electrochemical characterization of the deposition has shown that hydrogen evolution interferes with the Co-Ni co-deposition process. On the surface of a film, it is found that the H_2_ bubbles block the direct contact of the electrode with the electrolyte while the H_2_ bubbles can interrupt the growth of nanowires, resulting in fragmented nanowires. 

Morphology: the film deposited was relatively smooth compared to overgrown nanowires with a mushroom-like morphology. The mechanism of film formation is different for film and overgrown nanowires. The overgrown nanowires will eventually adopt the shape of a film that continues to grow by an electrochemical Ostwald ripening mechanism.

Composition: the effect of anomalous deposition was illustrated in the overgrown nanowire films. While the films were characterized by large areas with uniform composition, the overgrown nanowire arrays showed localized non-uniform composition. The mushroom cups of the overgrown nanowires had a composition rich in Co while the valleys in between them were rich in Ni.

Ni content: neither film nor nanowire composition is linear with the deposition potential. Co-Ni thin films and nanowires grown at −1.1 V had the highest Ni concentration of 73% and 76%, respectively, which is the highest content observed for Ni-rich deposit in a single bath. We remarked that a Ni/Co ratio of 3 was achieved only for nanowires, while Ni/Co ~2.5 was obtained for film. 

## Figures and Tables

**Figure 1 nanomaterials-09-01446-f001:**
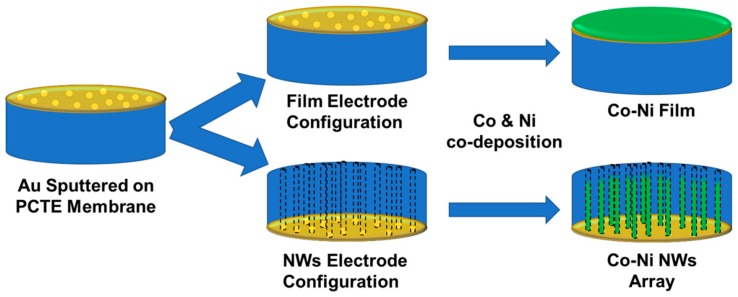
Illustration of the sample setup in a film electrode configuration (Au film faces the electrolyte) and in a NW (nanowire) configuration (Au film faces down to expose the pores to the electrolyte).

**Figure 2 nanomaterials-09-01446-f002:**
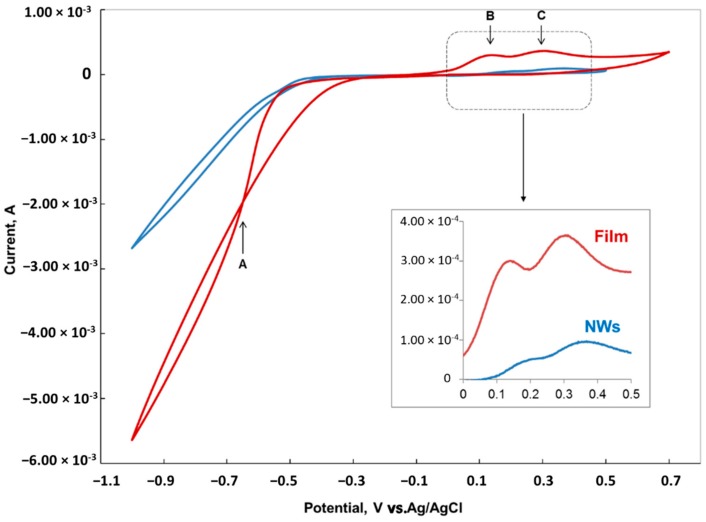
Cyclic voltammetry of Au in the Co-Ni solution at 20 mV/s sweep rate (red and blue lines are the scans for film and NW configuration electrodes, respectively). The inlet shows a magnified portion of the cyclic voltammetry (CV) in the potential range from 0 to 0.5 V.

**Figure 3 nanomaterials-09-01446-f003:**
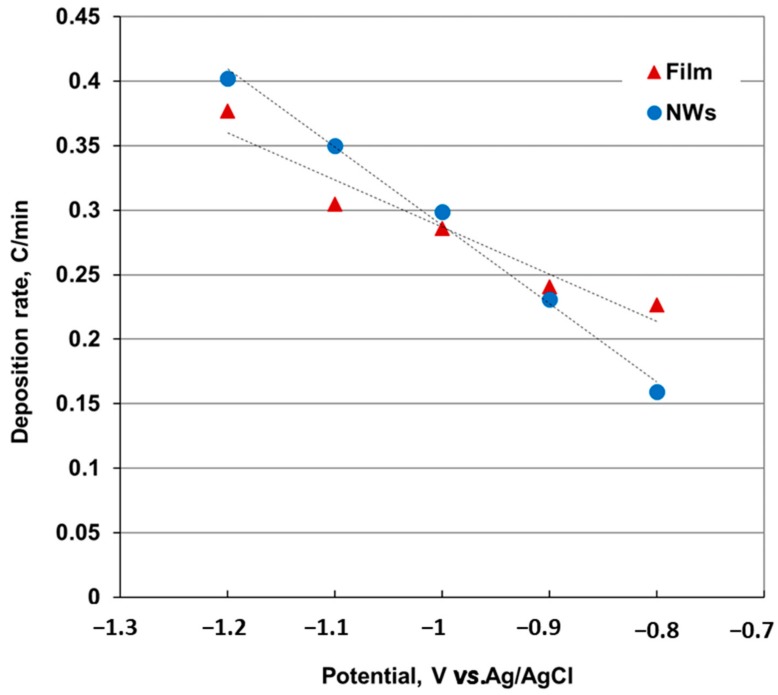
Comparison of the Co-Ni deposition rate for thin films and nanowires at different deposition potentials (the dotted line indicates the linear trend and is shown only to visually guide the variation of the deposition speed with the applied potential).

**Figure 4 nanomaterials-09-01446-f004:**
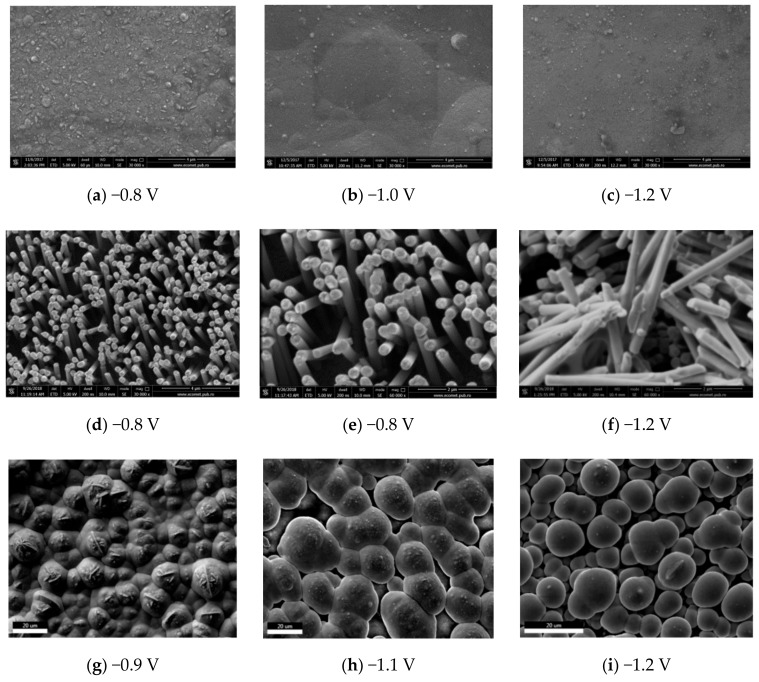
Scanning electron microscopy (SEM) images of Co-Ni film deposited on Au for 15 min (**a**–**c**), Co-Ni NW array deposited on Au for 5 min (**d**–**f**) and Co-Ni overgrown NWs deposited for 15 min (**g**–**i**) nanowire array: (**a**) perspective view of the NWs array deposited at −0.8V (×30,000); (**b**) NWs deposited at −0.8 V (×60,000); (**c**) NWs deposited at −1.2 V (×60,000).

**Figure 5 nanomaterials-09-01446-f005:**
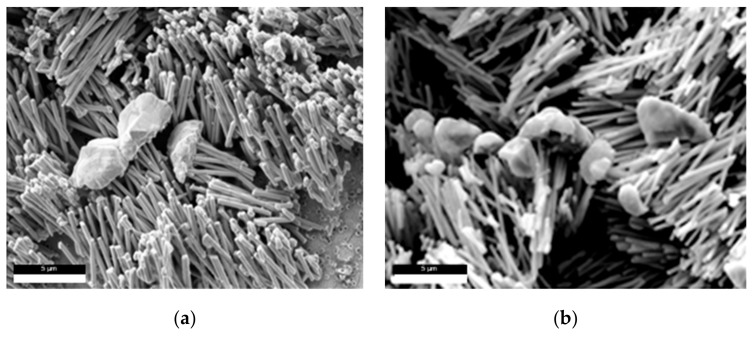
SEM images of Co-Ni nanowire array deposited at −1.2 V for 3 min (**a**) and 5 min (**b**).

**Figure 6 nanomaterials-09-01446-f006:**
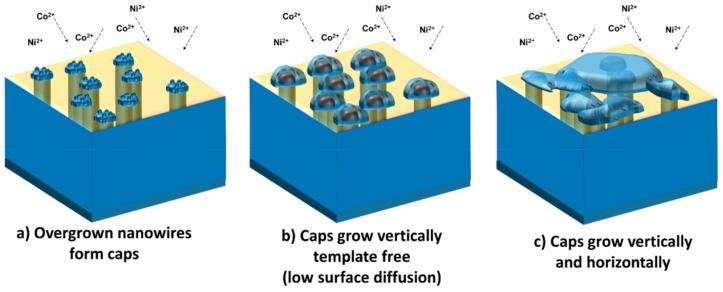
Illustration showing the growth of nanowire “mushroom” caps after the nanowires reach the surface. The membrane surface is illustrated here as a transparent area to better see the nanowires embedded in the membrane.

**Figure 7 nanomaterials-09-01446-f007:**
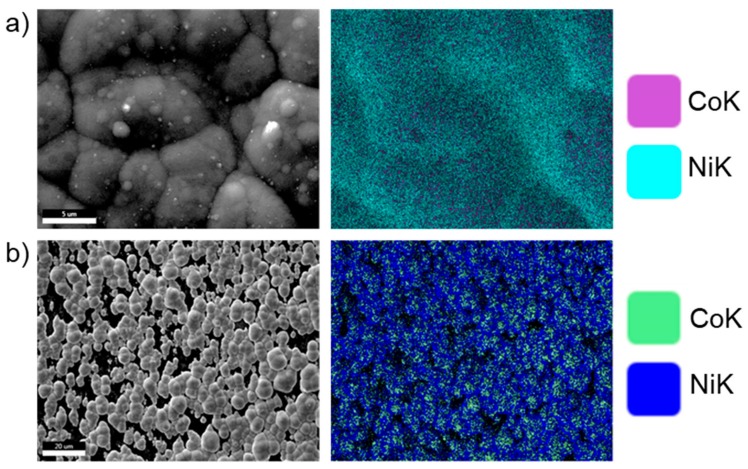
SEM image (**left**) and elemental mapping (**right**) of the Co-Ni films obtained at −1.0 V for 15 min: (**a**) on film and (**b**) overgrown nanowires.

**Figure 8 nanomaterials-09-01446-f008:**
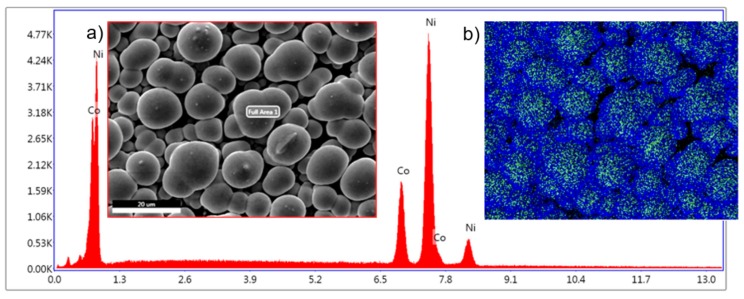
Illustration showing the growth of nanowire “mushroom” caps after the nanowires reached the surface. The membrane surface is illustrated here as a transparent area to better see the nanowires embedded in the membrane. SEM image of analyzed area (**a**) and EDAX color map showing the presence of both Co and Ni in the structures (**b**)

**Figure 9 nanomaterials-09-01446-f009:**
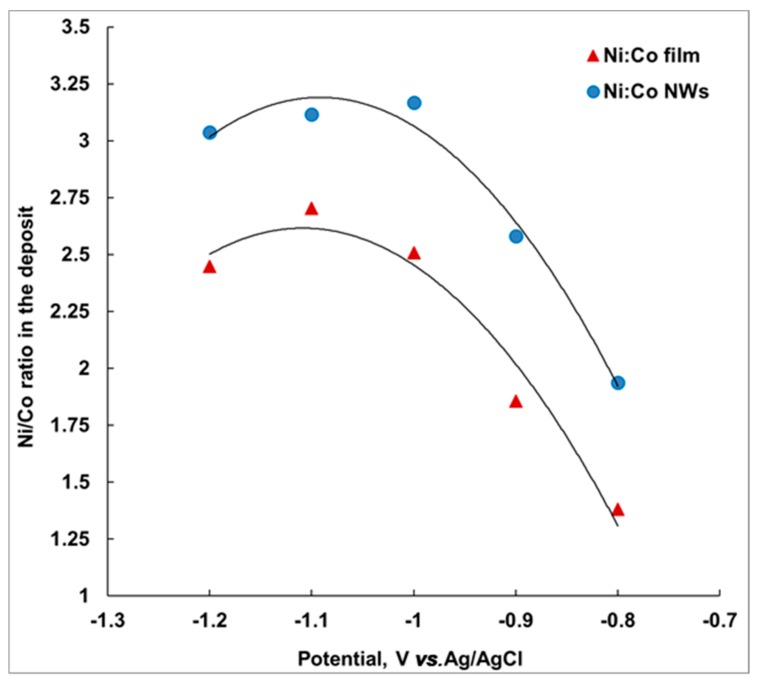
Potential dependence of the Ni/Co ratio in the deposit for film and nanowires at different potentials (the solid lines indicate the polynomial trend and are shown only to visually guide the variation of the Ni/Co ratio with the applied potential).

**Table 1 nanomaterials-09-01446-t001:** Composition of the deposits obtained at different applied potentials.

E, V	Film		Nanowires	
	at.% Co	at.% Ni	at.% Co	at.% Ni
−0.8	42.00	58.00	34.05	65.95
−0.9	35.00	65.00	27.93	72.07
−1	28.50	71.50	24.00	76.00
−1.1	27.00	73.00	24.30	75.70
−1.2	29.00	71.00	24.77	75.23

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
