# Peer review of "Template-Assisted Co-Ni Nanowire Arrays"

_nanomaterials, 2019, doi:10.3390/nano9101446_

Round 1

Reviewer 1 Report

The manuscript describes the study of Co-Ni nanostructures (thin layers and nanowires) obtained by electrochemical co-deposition. The thorough investigation of structural and electrochemical properties of produced nanostructures was performed. Authors obtained very interesting findings relating to the synthesis of Co-Ni nanocomposites. In my opinion, the manuscript could be accepted after minor revisions (the quality of formulas should be improved; there are some typos)

Author Response

We thank reviewers for their thoughtful comments on the original version of the manuscript.

We have modified the manuscript accordingly (changes are marked in red) and improved the quality of the formula (see Line 158).

Reviewer 2 Report

This is an interesting paper, having read it, the paper gives SEM/EDAX data, but no further structural or magnetic data,both of which would add to the paper and likely to change between the film and the nanowire, plus the different potentials. This would make the paper a lot stronger and be more useful for a wider audience.

minor suggestions and corrections include:

pg 3 line 102 define SHE eqn 1 seems to be in a funny format Fig 3 in the caption explain what the dashed lines are error on the data presented  Fig 4 caption has two different labels for a, b and c can this be checked, as the last ones don’t seem to correspond to the images fig 5 caption suggests a - I figures but there are only 2 images, please correct pg 9 line  308 sentence “which can be seen the morphology of the film” does not make sense, please rewrite Line 309 “nanosized plate” add a number in here to demonstrate this line 310 “ the morphology of Co deposits” does not make sense please rewrite figure 9 in the caption explain what the lines represent, also what is the error on these points line 348 starting “Ni content ....” end of this sentence does not make sense, please correct   

Author Response

We thank reviewers for their thoughtful comments on the original version of the manuscript.

We have modified the manuscript accordingly and all the suggested corrections have been addressed (changes are marked in red). See changes in the following lines: L102; L159; L168-169; L221; L307; L309; L347-348;

Reviewer 3 Report

Manuscript number: nanomaterials-602522

Manuscript Title:

 Template-Assisted Co-Ni Nanowire Arrays

Overall Evaluation

The manuscript describe an interesting  comparison between Co-Ni thin films and template-assisted nanowires  arrays obtained by electrochemical co-deposition. Different techniques have been used in order to give an exhaustive characterization.

This paper can be accepted after minor revision.

Abstract and Introduction

Page 1, line 14/15: I suggest to change the sentence “was performed at constant potentials, from E = -0.8 to 14 -1.2 V vs Ag/AgCl” in “was performed at constant potentials chosen in the range from E = -0.8 to 14 -1.2 V vs Ag/AgCl”. Page 2, line 85 there is a typos in cm2.

References

Page 12, line 397, 415, 424, in the references 5, 12 and 15 the page numbers are missing.

..

Author Response

We thank reviewers for their thoughtful comments on the original version of the manuscript. We have modified the manuscript accordingly and all the suggested corrections have been addressed (changes are marked in red). See changes in the following lines: L14-15; L85.

Regarding the references, we used Endnote to import papers citation. We checked the references and indeed, the page is missing. Investigating why the EndNote did not include the page number in the reference, we found out that this is due to the change in MDPI Journals policy to list the article number in place of the article page. See for reference the following notification that appears on the MDPI page at: https://www.mdpi.com/about/announcements/784: "From the January 2016 issue, MDPI journals will use article numbers in place of the traditional method of continuous pagination through the volume. This step helps us to maintain a rapid, efficient production process by being able to define pagination as soon as a paper is accepted. We have recently made a number of improvements to article layout and production procedures, of which article numbers is just one component. You can read more about the changes at http://blog.mdpi.com/2015/12/01/a-new-look-for-mdpi-papers/. For papers that use article numbers the page number will start from 1 and the citation needs only list the article number, e.g., Holmes, L.; LaHurd, A.; Wasson, E.; McClarin, L.; Dabney, K. Racial and Ethnic Heterogeneity in the Association Between Total Cholesterol and Pediatric Obesity. Int. J. Environ. Res. Public Health 2016, 13, 19 (article number: 19. https://www.mdpi.com/1660-4601/13/1/19).” Therefore, we added the article numbers to the following citations: 5, 12, 15.

Round 2

Reviewer 2 Report

The authors have corrected the minor errors in the paper, I would have liked a comment on why there is no magnetic or XRD data by the authors as originally requested.